# Biomechanical Characteristics and Analysis Approaches of Bone and Bone Substitute Materials

**DOI:** 10.3390/jfb14040212

**Published:** 2023-04-11

**Authors:** Yumiao Niu, Tianming Du, Youjun Liu

**Affiliations:** Faculty of Environment and Life, Beijing University of Technology, Beijing 100124, China

**Keywords:** bone, biomaterial, collagen, mineralization, biomechanics

## Abstract

Bone has a special structure that is both stiff and elastic, and the composition of bone confers it with an exceptional mechanical property. However, bone substitute materials that are made of the same hydroxyapatite (HA) and collagen do not offer the same mechanical properties. It is important for bionic bone preparation to understand the structure of bone and the mineralization process and factors. In this paper, the research on the mineralization of collagen is reviewed in terms of the mechanical properties in recent years. Firstly, the structure and mechanical properties of bone are analyzed, and the differences of bone in different parts are described. Then, different scaffolds for bone repair are suggested considering bone repair sites. Mineralized collagen seems to be a better option for new composite scaffolds. Last, the paper introduces the most common method to prepare mineralized collagen and summarizes the factors influencing collagen mineralization and methods to analyze its mechanical properties. In conclusion, mineralized collagen is thought to be an ideal bone substitute material because it promotes faster development. Among the factors that promote collagen mineralization, more attention should be given to the mechanical loading factors of bone.

## 1. Introduction

Bone is a stiff, strong, and tough organ that serves as a vital supporting organ in the body. It is composed of a hierarchically structured and naturally optimized bone matrix. Furthermore, it has an abundance of blood vessels and nerves that can constantly carry out metabolism, growth, and development, as well as reconstruct, repair, and regenerate [1]. The human body contains a total of 206 bones. Except for the six auditory ossicles that belong to the receptors, bones are classified according to their location as the skull, vertebrae, and limb bones [2]. Bone is classified into four types based on its morphological characteristics: flat bone (such as the spine), long bone (such as the humerus, femur, etc.), short bone (such as the carpal bone), and irregular bone (such as plate scapula). The long bones provide structural support for the motor system and support body movement, whereas the flat, short, and irregular bones can fill and protect the body (such as the skull) and help the body complete life activities more flexibly and efficiently (such as sesamoid bone) [2].

### 1.1. Composition and Structure of Natural Bone

Bone tissue is a complex structure composed of inorganic and organic matter, making it one of the most complex compounds in nature. It is primarily composed of inorganic (65%) and organic phases (30%) [3]. The perfect combination of organic and inorganic materials gives the bones good stiffness and toughness. The inorganic phase is composed of calcium phosphate, primarily HA (HA, Ca_10_(PO_4_)_6_(OH)_2_), and is stiff and strong, making it an ideal carrier for mineralized collagen [4]. Type I collagen is the main organic component of the bone matrix. Osteoblasts are the cells that produce collagen. The general process is as follows: first, the collagen polypeptide helical chain is synthesized, and then the peptide chain is modified by amino acid (proline and lysine) hydroxylation to self-assemble the triple helix to form collagen fibrils [5]. Individual collagen fibrils are approximately 1.6 nm in diameter and 300 nm in length [6]. Collagen fiber mineralization begins when the body synthesizes collagen fibrils; that is, the collagen fibers are forming in a periodic structure. While collagen fibers assemble, the inside and outside gaps are filled with HA nanocrystals, and mineralized collagen is formed [7]. These mineralized collagen fibers serve as the foundation for cortical and cancellous bone [8].

The structure of natural bone tissue has a multiscale and multilevel range from micro to macro. Weiner and Wagner first proposed a seven-level hierarchical structure of bone tissue by studying the femur [9]. From macro to micro, the sequence is as follows: the whole bone tissue, cancellous and dense bone, cylindrical Haversian canal (bone unit), the parallel or staggered arrangement of mineralized collagen fiber bundles, mineralized collagen fiber bundles, micron-scale mineralized collagen fibers, and nanoscale HA and collagen molecules [9]. Procollagen microfibers are the smallest unit of bone tissue composition, and it is assembled in an orderly hierarchical manner to form a macroscopic bone in general. Reznikov et al. further classified bone tissue into a nine-level hierarchical structure based on this [10]. The previously proposed hierarchical structure theory was expanded to include structures at the histological hierarchy (100 nm) and lamellar bone structures (10 µm). They then revealed the crystal morphology and orientation patterns by extracting slices of the lower femoral neck, using scanning transmission electron microscopy and three-dimensional (3D) reconstruction and electron diffraction, and combing them with crystallographic data to establish the corresponding model and broaden the structure of the bone to 12 layers (Figure 1) [11].

### 1.2. Mechanical Properties of Natural Bone

Bone possesses the exceptional properties of both collagen and HA, namely rigidity and toughness [12], making it an ideal structural material for the human body that is light but strong. Numerous studies have revealed that bone strength is affected not only by its composition but also by bone mass, geometry, and microstructure. The anisotropic behavior of bone materials and the magnitude of stress intensity vary slightly across the bone [13]. On the microscopic level, the needle-shaped inorganic salt crystals are primarily arranged longitudinally along the collagen fibrils, whose primary function is to connect and constrain the longitudinal fibers so that they are not unstable under compressive and bending loads [14]. Collagen, on the other hand, binds to inorganic salt crystals, and collagen is a common biopolymer that can provide toughness to biologically hard tissue materials [15]. The hollow beam structure of bone can greatly improve the bending strength without increasing the weight [16,17] (Figure 2A). Furthermore, the internal organization of the bone demonstrates that it is a reasonable load-bearing structure. According to the comprehensive stress analysis, the area that bears high stress also has high strength. The arrangement of femoral trabeculae, for example, is very similar to the stress distribution. To withstand greater stress, materials with higher density and strength are arranged in the internal structure of bone in the high-stress area [18].

It is an anisotropic and uneven bone composite material, and its mechanical properties are evidently different individually and by parts, as is the hardness of bones in different parts. As one of the most important properties of bone, bone hardness includes elastic deformation and plastic deformation. The nanoindentation method was used to measure the hardness of human bones, which provided valuable data for the preparation of bone repair materials (Table 1) [19,20,21,22,23].

**Figure 2 jfb-14-00212-f002:**
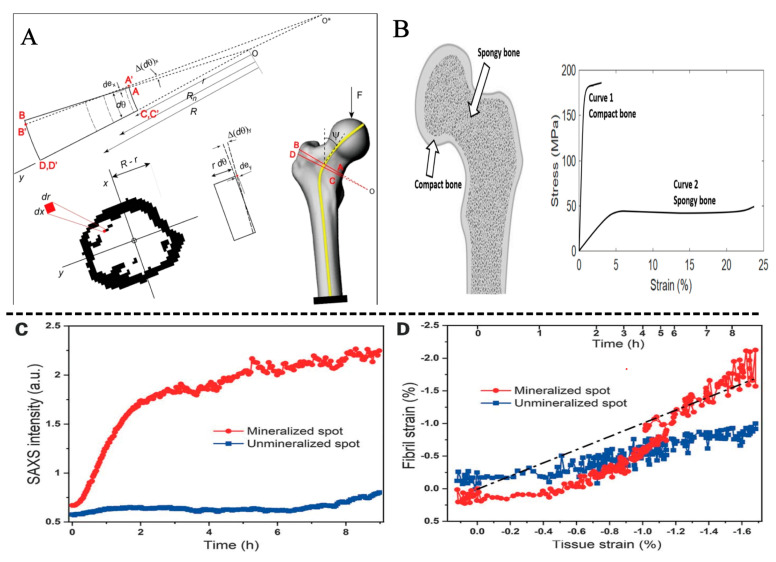
(**A**) Representation of the curve beam model of bone, Reprinted with permission from Ref. [17] (2023, Springer Nature). (**B**) Bone can be divided into two types: cortical bone and cancellous bone. Right panel: Typical stress-strain curves of cortical and cancellous bone, Reprinted with permission from Ref [24] (2023, JMNI). (**C**) Integrated small angle X-ray scattering (SAXS) intensity of mineralized and unmineralized regions. (**D**) Correlation between tissue and fibril strain in the mineralized and unmineralized spots. (**E**) Image of tendon after 4 h of mineralization. (**F**) Schematic of the evolution of strains during tendon mineralization. In the mineralized area, collagen fibers shrink Reprinted with permission from Ref. [25] (2023, AAAS).

The bone has several irregular marrow cavities due to its structure. Bone is classified into two types based on its size and density: cortical (dense) and cancellous (spongy) (Figure 2B). The proportion of each bone varies; however, on average, cortical and cancellous bones account for approximately 80% and 20% of the bone, respectively. These two skeletal components are identical, but macroscopic and microscopic structures differ [24]. The cortical bone serves as the shell of the entire skeleton. The gap within cortical bone is much smaller. The cortical bone has a porosity of 5–10% and an apparent density of 1.5–1.8 g/cm^3^ [26]. Cancellous bone is found at the end of the bone or within it, surrounded by outer cortical bone. Cancellous bone consists of thin columns called trabeculae that are loose and dense, with porosity of 50–90% and an apparent density of 0.5–1.0 g/cm^3^ [27]. Porosity is one of the most crucial factors that affect the mechanical properties of bone. As a result of significant differences in porosity, the mechanical properties of cortical and cancellous bones are significant. Cortical bone can be tolerant of higher stress (approximately 150 MPa) and lower strain (approximately 3%) before failure, and cancellous bone can be tolerant of lower stress (approximately 50 MPa) and higher strain (approximately 50%) before failure [24]. Furthermore, the distribution of cortical and cancellous bones in the body varies. Cancellous bone is commonly found in the long bone metaphysis, vertebral body, and the interior of the pelvis. By contrast, cortical bone is lamellar and commonly found on the surface of the long bone diaphysis and cancellous bone (such as the vertebral body and pelvis). Furthermore, collagen fibrils are mineralized with HA during bone formation. Mineral precipitation has been shown in studies to cause collagen fibril contraction of collagen fibrils at stress levels of several megapascals. The dimension of the stress depends on the type and quantity of mineral [25].

## 2. Biomechanical Properties of Biomimetic Bone Materials

In recent decades, bone tissue engineering has received great attention because of its potential to repair the bone matrix of traumatic or nontraumatic destruction. However, because of the different contents of cortical bone and cancellous bone, the biomechanical properties of different parts and shapes of bone are also quite different [24]. In this study, we briefly describe the common types of bone repair and the scaffolds for bone repair.

### 2.1. Common Types of Bone Repair

Bone differentiates into various shapes and structures based on its roles and functions. Correspondingly, the contents of cortical and cancellous bones vary depending on their location and shape [24]. This indicates that the biomechanical properties of these bones are quite different because of their different porosities [28,29]. There are several classification methods for bone in the academic world. In this study, we divided human bone into load-bearing bone and non-weight-bearing bones according to the location and load size of the bone. Load-bearing bones bear most of the load of the human body, including mainly the spine, limb bones, and joints, whereas non-weight-bearing bones include mainly the skull, maxillofacial, orbital bones, and ear ossicles.

The knee joint bears the maximum joint pressure in daily life, which is approximately 4–4.5 times the body weight [29]. Furthermore, when the body walks, this multiplies further [30]. The hip joint, ankle joint, wrist joint, and other load-bearing parts are subjected to a great deal of stress [31]. When the skeleton is damaged, such as a common bone disease osteoporosis, bone with this condition will become very fragile and prone to fracture, especially in weight-bearing areas such as the pelvis, hips, wrists, and spine [32]. This implies that scaffolds with a similar strength to the original bone need to be designed and that when considering biomimetic alternatives for these parts, materials with good mechanical properties must be chosen.

Compared with load-bearing bone, non-weight-bearing bone is subjected to less mechanical load and has more roles in filling and protection [29]. In recent years, because of the development of medical aesthetics and dentistry, some non-weight-bearing bone replacement materials have received extensive attention (Table 1). Its application can be roughly divided into two parts: non-weight-bearing bone orthopedic implants and bone defect filling. Non-weight-bearing bone orthopedic implants are mainly used in orbital implantation, ossicular replacement, and nasal bone injury. In addition, the main methods of bone defect filling are alveolar ridge elevation, tooth replacement, and maxillofacial reconstruction. For bone defect repair, we summarized different reference repair materials for different sites of the bone defect (Table 1). According to the different application scenarios of restorative materials, orthopedic implants can be divided into non-weight-bearing and load-bearing implants.

**Table 1 jfb-14-00212-t001:** Application of bone repair materials in common sites.

	Repair Site	Vickers Hardness [19,20,21,22,23]	Material Properties	Application Features	Example
Load bearing bone	Limb bones	40.39–44.59 HV	Metals and Alloys	Weight-bearing, Correction, Immobilization	Nano-titanium and Ti-6Al-4V alloy [29]
	Joints	38.55 HV			
	Spine	25.47–32.80 HV			
	Ribs	37.35 HV			
	Skull	39.86 HV			
Non-weight-bearing bone	Maxillofacial	43.54 HV	Bioceramics	Fill, Support, Protect	Calcium Phosphate, HA [33,34,35]
	Orbital	42.95 HV			
	Dental	278–285 HV			
	Middle ear bone	54.11 HV			
	Cartilage	0.317 HB	Polymer	Fill, patch, join, join	Collagen and PLA nanofibers [36,37]
	Maxillofacial	43.54 HV	Composite material	Fill, repair	HA-Collagen [38,39,40,41]
	Dental	278–285 HV			

### 2.2. Load Bearing Implant

Load-bearing implants are artificial knee joint and hip joint prostheses and intervertebral fusion, which are used for the limbs and trunk of the implants. These implants do not only have the effect of filling defects but also need to bear the weight and load in the process of movement of patients; therefore, they need higher mechanical properties [42]. Metal (Figure 3) has become the preferred material for load-bearing implants because of its excellent mechanical characteristic and ability to withstand physiological loads. Typically, these materials are stainless steel, cobalt-chromium (Co-Cr) alloy, titanium (Ti), and Ti alloy [43,44]. Although Co-Cr alloy has excellent corrosion resistance, its friction property is poor, which limits its application in joint prostheses. Of all these metals, Ti and its alloys are the most resistant to corrosion [45,46]. Several Ti alloys, such as Ti-6Al-4V and Ti-6Al-7Nb, have sufficient strength and corrosion resistance [47]. However, its main drawbacks are its high cost, poor wear performance, oxygen diffusion to Ti during manufacturing and heat treatment, and dissolved oxygen, which causes Ti embrittlement [48]. In addition, some problems are inevitable. The difference in Young’s modulus between metallic materials and bone induces changes in the mechanical stress field, leading to adaptive remodeling and a decrease in local bone density [49,50]. Moreover, the adverse effects of metal materials implanted in the human body need to be reduced caused by fatigue fracture, corrosion, and metal corrosion [51].

Researchers have created porous scaffolds to improve scaffold performance by reducing the influence of stress shielding in metals and alloys [54]. The final density, pore size, material type, and preparation parameters all significantly impact the mechanical properties of porous scaffolds [45]. For example, when the porosity increased from 55% to 75%, the strength of the spongiform bone-like biomimetic Ti scaffold decreased from 120 MPa to 35 MPa [55]. In general, with increasing porosity, the effect of stress shielding is gradually weakened, and it is more conducive to the growth of cells between tissues. However, although high porosity can provide space for bone growth, which is conducive to implant fixation, with increasing porosity, the strength and extensibility of porous structure will decrease; therefore, the porosity also needs to be controlled within a certain range [56]. It is necessary to control the porosity and pore size of the scaffold accurately. Among the several methods, 3D printing has attracted much attention because of its excellent properties, which designs scaffolds with not only different shapes and sizes but also different pore percentages and mechanical strengths [57].

### 2.3. Non-Weight-Bearing Bone Implants

Non-weight-bearing bone implants are mainly internal fixation implants such as bone plates and bone screws, and filling implants are applied to repair bone defects in non-weight-bearing areas [58]. These implants are used for structural fixation and filling but generally are not used for load bearing. In addition to metal materials for some internal fixation implants, degradable polymer and ceramic materials with similar inorganic composition to bone are preferred materials for non-weight-bearing bone implants [38,39]. Polymers, bioceramics, and composite materials can be classified based on their chemical structures.

Polymers can control the shape, structure, and chemical composition of materials and can be used to fabricate bioscaffolds as artificial biomaterials. Polymeric biomaterials are typically implanted into the human body in various forms, such as tissue scaffolds, gels, particles, or films and degraded into non-toxic products that are absorbed or excreted by the human body through enzymatic reactions [40]. Synthetic polymers, such as poly (a-hydroxy acid), are degraded in vivo to non-toxic lactic acid and glycolic acid, which can be eliminated from the human body by through normal excretion [41,59]. Although synthetic polymer materials are relatively easy to process into a pore scaffold, they may also produce acidic degradation products and change the pH around the tissue. This change in pH affects cell behavior and survival and causes tissue inflammation [60]. Natural biological materials do not have the problems that polymers do, and they have excellent biological activity, biocompatibility, and controllable degradation, all of which are crucial components of tissue engineering materials [61]. Naturally derived biomaterials are typically divided into two categories: protein-based biomaterials, such as collagen and sericin, and glycosyl biomaterials, such as hyaluronic acid and cellulose [62]. However, the degradation rate of naturally derived biomaterials in vivo is not only difficult to control and anticipate, but the mechanical properties are also weak, and the uniformity of composition cannot be regulated [53].

Ceramic material is a type of biological material with a crystal or partially crystal structure, which is stiff but fragile [63]. Moreover, its mechanical properties are associated with chemical elements. Generally, the chemical elements used to make bioceramics are only a small part of the periodic table, indicating that bioceramics can only be made of alumina, zirconia, carbon, and silicon- and calcium-containing compounds [64]. Therefore, bioceramics have excellent biological functions and biocompatibility. For example, after implantation, the formation of apatite on the surface of bioceramics makes the combination of internal tissues and implants stronger [65]. However, its mechanical properties are influenced by its elements and structure, making it stiff and fragile correspondingly [64]. Researchers frequently consider incorporating biological active agents as composite materials.

There are many kinds of chemical elements in the human body. They interact with each other and maintain life homeostasis together. A variety of metal elements, such as magnesium (Mg), zinc (Zn), manganese (Mn), strontium (Sr), copper (Cu), cobalt (Co), ferrum (Fe), aluminium (Al), nickel (Ni), and chromium (Cr), have been found to induce proliferation during tissue regeneration [66]. They also play an important role in promoting bone biomineralization [67]. Therefore, the introduction of metal elements can not only improve the mechanical properties of HA bioceramics but also promote the proliferation, differentiation, and migration of active cells in the bone to regulate bone mineralization. The incorporation of magnesium into bioceramics can promote bone proliferation [68]. The introduction of Hap into Fe can increase biocompatibility and blood compatibility [69]. In the biological experiment of β-SiAlON ceramics, the cells cultured on the surface of β-SiAlON were observed. The increase in AlO_2_ concentration had no effect on cell adhesion and spreading, but it may slightly inhibit cell proliferation at high concentrations. Low AlO_2_ concentration helps to promote osteogenic differentiation and mineralized nodule formation [70]. Zn-containing bioceramic scaffolds in craniofacial bone repair experiments show that soluble Zn^2+^ can promote osteogenic differentiation of adipose stem cells [71]. Boron silicate nanoparticles merged with Cu and Mn can be used for fusion bone repair and anti-tumor therapy. It can enhance bone regeneration through the osteogenesis of Cu^2 +^ and Mn^3 +^ and induce tumor cell apoptosis through Cu^2+^ and Mn^3+^ [72]. Sr is a trace element in the human body that is beneficial to bone formation [73]. Sr has a strong affinity for bone. Due to the physical and chemical similarity with Ca, the interaction of Sr in bone tissue is similar to that of Ca [74]. Sr can inhibit the osteoclast differentiation of pre-osteoclast cells and promote the expression of outcome cells and protein secretion. The subsequent rabbit bone scaffold implantation experiment also proved this. The Sr-doped scaffold provides a suitable environment for cell proliferation and differentiation during degradation [75,76]. In addition, the doping of some rare metals, such as praseodymium (Pr), erbium (Er), and yttrium (Y), can also promote cell proliferation and differentiation [77,78,79].

Composites are typically made up of polymers and mineral salts, with the mineral phase primarily consisting of phosphate, silicate, and other minerals [80]. Composite materials combine polymer toughness and mineral hardness of minerals and become the first choice for future biological materials. For example, Bogdan Conrad and Fan Yang prepared scaffolds from HA-mineralized gelatin, whereas Chen et al. used HA-mineralized silk fibroin (SF)/cellulose [81,82]. 

Bioglass incorporation into collagen scaffold as a relatively broad composite material has attracted much attention due to its excellent degradability and stability. Bioglass is an excellent biomaterial which is often used in bone defects. Before this, people often combined polylactide-co-glycolide with organic glass to improve the mechanical properties of composite scaffolds [83,84]. Collagen has become the preferred matrix for bioglass doping among many biodegradable materials due to its excellent biocompatibility and biological temperature [85]. In contrast, the incorporation of inorganic bioactive glass has been shown to increase biological activity and mineralization, control scaffold degradation, and improve the mechanical properties of collagen scaffolds [86,87]. Existing studies show that in vivo, mineralized scaffolds doped with bioglass can promote the mineralization of collagen. Nijsure et al. successfully prepared bioglass-incorporated electrochemically aligned collagen [88]. The incorporation of bioglass-incorporated electrochemically aligned collagen will enhance the mechanical properties and cell-mediated mineralization [88]. The dissolution product of the bioglass collagen composite scaffold stimulates osteoblast differentiation and extracellular matrix mineralization in vitro without any osteogenic supplement [89].

As an emerging option for composite materials, biomimetic mineralized collagen is a highly mineralized composite material composed of collagen and HA. It has attracted much attention because it has the same composition as the bone matrix. In addition, their mechanical properties and microstructure are similar to the extracellular matrix of native tissues [90]. Because of its exceptional biological activity, osseointegration, and biological induction ability, it is widely favored by researchers. Wang Xiumei’s team developed high-strength bone materials mimicking compact bone and completed the skull defect experiment of adult ovis aries. The results showed excellent osseointegration and osteogenic induction abilities [91,92]. In recent years, mineralized collagen-guided bone tissue regeneration has gradually been used in oral clinical treatment, primarily for the treatment of bone and soft tissue defects caused by periodontal disease or cysts [93,94,95]. Moreover, while mineralized collagen made good progress in repairing other bone defects, it has the following limitations: the implant material lacks structural strength and requires external fixation increasing patients’ pain. Notably, biomimetic mineralized collagen materials, like other traditional bone repair materials, have not been widely used in clinical practice due to insufficient mechanical strength [96]. How to improve mechanical strength is also a research focus.

## 3. Study to Improve the Mechanical Strength of Mineralized Collagen

Among the various methods for improving the mechanical properties of collagen, in vitro biomimetic mineralization is an effective method for achieving the most accurate simulation of natural bone tissue structure. According to the principle of biomimetics and the metabolism law of human tissue, HA/collagen composite material construct not only has the macroscopic structure of natural bone but also simulates its microscopic characteristics, revealing the benefits of HA and collagen materials complementing each other and significantly improving the compression modulus of the composite scaffold added with HA [97]. It has excellent biocompatibility, bone conductivity, and osteoinductive ability [98]. Numerous studies have been conducted to promote the collagen mineralization process more effectively and enhance the degree of collagen mineralization. At present, the factors affecting the in vitro biomimetic mineralization of collagen include mechanical, biological, chemical, and collagen structure.

### 3.1. Force to Promote the Mineralization of Collagen

The process of bone healing is affected by several factors, such as biological, chemical, and mechanical factors. Several studies have demonstrated that force acts directly on bone matrix and then on cells. Bone remodeling requires the participation of osteoblasts and osteoclasts. Osteoblasts and osteoclasts respond to force stimulation and show different proliferation abilities and activities. During bone remodeling, mechanical force stimulates the fracture site, accelerating bone formation and inhibiting bone resorption [99]. In contrast, when the body is not stimulated by force for a long time (such as bed rest, joint fixation after surgery, or exposure to a microgravity environment), the body will lose more skeleton. In severe cases, osteoporosis will occur [100,101,102].

Various methods for simulating the force environment of osteoclasts and osteoblasts in the bone matrix, including fluid shear stress, cyclic stretching, continuous compression force, and mechanical stress from liquid perfusion or compressed air, have been developed to investigate cellular responses to mechanical stimuli [103,104,105,106,107,108,109,110]. Physiological mechanical loading enhances the antiapoptotic effect and promotes osteoblast proliferation and differentiation, resulting in extracellular matrix formation and bone remodeling [106,107,108].

In addition, stress can induce the mineralization process of collagen. On the one hand, stress can affect the self-assembly of collagen, and on the other hand, stress can also induce collagen mineralization. The experiments were performed within a microfluidic channel, and the size of the channel affects the angular size of the collagen alignment with the axis of the microfluidic channel. Collagen fiber alignment decreases with increasing channel size [109]. Du et al. used a cone-and-plate viscometer to provide fluid shear stress [110]. The results showed that the formation of amorphous calcium phosphate (ACP) was associated with its rate. Fluid shear stress can significantly affect the ACP by somatic transformation and the crystal structure of HA transformed from precursors. Subsequently, periodic shear stress was used again to induce collagen mineralization. The results showed that periodic fluid shear stress could control the size of ACP, such as polyacrylic acid (PAA), avoid aggregation, and contribute to the formation of intrafibrillar mineralization (Figure 4A,B) [111]. Cyclic tensile experiments on demineralized bone demonstrated that cyclic strain increased the migration of mineralized fluid with mineralized precursors to the matrix, resulting in the formation of more calcium phosphate nanocrystals and an increase in the elastic modulus of the collagen matrix (Figure 4C,D) [112]. However, when a constant tensile force is added to the demineralized bone, the mineralization of the demineralized bone is also inhibited. The experiments of Clinical Dentistry showed that collagen mineralization could be more effectively induced by the flowing mineralization solution under focused high-intensity ultrasound. In addition, the amount of mineral formation is proportional to the exposure time [113].

### 3.2. Collagen Fiber Arrangement Affects Mineralization

The bone structure is constantly regulated by the mechanical environment during the reorganization, thereby maintaining the mechanical strength. Bone is the basic structural unit of cortical bone, which is composed of a concentric lamellar structure around the central Haversian tube. Although the direction of the bone process is mainly parallel to the long axis of the bone, the direction of collagen fibers in a single layer may vary greatly, resulting in many models [10,115] proposed over the years. Over the years, researchers have also studied the effect of collagen fiber orientation on the mechanical properties of bone lamellar structure. The results also confirmed that collagen fiber orientation allows the bone to withstand greater stress without breaking [116,117,118]. Similarly, in the in vitro biomimetic mineralization experiment of collagen, it was also found that the arrangement of collagen fibers had a great influence on the formation and deposition of HA. As we mentioned earlier, type I collagen has a special amino acid sequence and triple helix structure. Cross-linking generates new chemical bonds through amino acids on adjacent peptide chains, which can improve the stability of collagen conception. In the body, cross-linking is an enzymatic or non-enzymatically mediated enzymatic process mediated by lysyl oxidase to produce trivalent collagen cross-linked pyridinoline (PYD) and deoxypyridino-line (DPD) [119]. In the body, in in vitro experiments, researchers often change the structure of mineralized collagen through physical or chemical cross-linking and then change the arrangement of collagen fibers [120], just as the existence of cross-linking makes the collagen structure different so that the mineralization of collagen is also different. Collagen with a different cross-linking degree was prepared by gamma-ray irradiation. With the increase of cross-linking degree, the pore structure of collagen became denser. The compact structure of collagen enables HA to adhere to collagen fibers [121]. However, using glutaraldehyde as a cross-linking agent to prepare mineralized collagen scaffolds, it was found that with the increase of cross-linking agent, the arrangement of HAP crystals in collagen fibers decreased, and improper use of cross-linking agent would inhibit the mineralization of collagen [122].

### 3.3. Other Methods to Promote the Mineralization of Collagen

In addition to in vitro biomineralization, other factors can promote collagen mineralization [111,123]. The use of a polymer-induced liquid precursor to mineralize collagen fibers can result in nanostructures that are extremely similar to the bone tissue matrix, according to nonclassical crystallization theory. Calcium ions gradually aggregate with phosphate in this process to form ACP, which is distributed inside and outside collagen fibers and converted to HA [124,125,126]. Polymers like polyvinyl phosphoric acid and PAA help in the formation of nanosized ACP [127]. As a biological small molecule, poly-aspartic acid can also control the formation of ACP, thereby achieving mineralization in collagen fibers [112]. In recent years, studies have also focused on the phosphorylation of collagen. Compared with the regulation of only orthophosphate, which can only form spherical mineralized crystals, needle-shaped mineralized crystals will be formed in the solution with the addition of alkaline phosphatase (ALP), thereby forming petal-shaped crystals on collagen (Figure 5A–F) [128]. In addition, subsequent experiments proved that compared with fluid shear stress alone, the pore density, hydrophilicity, enzymatic stability, and thermal stability of mineralized collagen were significantly improved after the addition of sodium tripolyphosphate [119]. Several experimental factors influence the size and distribution of HA nanocrystals in the pore region of collagen fibrils and among the fibrils by affecting the formation and transportation of ACP [129,130]. According to previous studies, collagen as the template for biomineralization, its fiber diameter, orientation, degree of cross-linking, and degree of phosphorylation can all affect mineralization. For example, the diameter of collagen affects the mineralization degree inside and outside of the fiber. A thicker collagen fiber is not conducive for HA to enter the fiber and mineralization inside the fiber [131]. With increasing cross-linking, the collagen will become denser, which is conducive to the deposition of HA and the production of highly mineralized collagen (Figure 5G–I) [121].

## 4. Method for Detecting Mechanical Properties of Mineralized Collagen

Although the mineralized collagen scaffolds have the same composition as bone, achieving similar structure and mechanical properties to that of natural bone has always been the focus and a challenge for researchers. In the preparation of mineralized collagen, the detection methods and standards are particularly crucial. Currently, researchers test the mechanical properties of collagen fibers, primarily from the macroscopic and microscopic perspectives, to analyze the material’s mechanical properties. This study concentrated on microscopic testing methods because macroscopic mechanical testing, or traditional mechanical testing, is relatively well-developed (Table 2).

### 4.1. Macroscopic Mechanics Analysis Methods

There are numerous methods to analyze the mechanical properties of materials. The traditional mechanical property testing techniques include stretching, bending, and torsion. Various testing methods improve material performance parameter acquisition methods and provide a broad avenue for material performance testing [132]. These traditional material testing techniques were performed earlier, and we summarized the research status of several typical testing methods in this study. The tensile test of materials can be divided into ex situ and in situ stretching based on real-time observation. Ex situ stretching is traditional stretching (Figure 6A,B). In addition, the branches are more complex and have several directions for development [133]. The studies conducted by people using the extensometer can measure not only the plastic deformation, elastic recovery, and tensile strength of the material but also the test temperature, load frequency, holding load, amplitude, and other parameters, and the total deformation of the specimen [134].

### 4.2. Microscopic Mechanics Analysis Methods

The continuous maturation of surface topography observation and internal structure flaw detection technology benefits the development of in situ stretching. Surface topography can be observed in situ using charged-coupled device cameras, optical microscopes, atomic force microscopy (AFM), scanning electron microscopy (SEM), and other instruments. The crystal structure of the material was studied using an X-ray diffractometer and a Raman spectrometer (Figure 6C,D) [132,135,136]. Micro-tensile, nanoindentation, and scratch tests are the main methods for detecting the microscopic morphology of materials in materials science. They can accurately measure the hardness and elastic modulus of materials. At the same time, with microscopic imaging, morphology changes can be observed and widely used. This part is further explained below.

#### 4.2.1. Micro Stretching

Micro stretching is an in situ stretching method based on the rapid development of optical microscopies, such as AFM, SEM, and other microscopic observation methods. Typically, micro-stretching can reflect the mechanical changes of the material on a micro- and nano-scale level. Among them, the combination of SEM and the tensile mechanical testing device is an earlier in situ method used in material studies [137]. The imaging speed of SEM is fast, and the micro- and nanoscale topography can be observed clearly. It can provide detailed information on the behavior of materials during mechanical testing that static observation cannot. Some studies performed in situ SEM mechanical tests on transverse and longitudinal bone specimens to further verify the anisotropy of bone mechanical properties and proposed that the mechanical properties of the longitudinal and transverse orientations of the bone were different, which could be attributed to differences in the direction of microcracks [138]. Furthermore, a study reported a novel device with a confocal Raman microscope that enables uniaxial stretching of microfibers ranging in diameter from 10 to 100 microns in length [135].

#### 4.2.2. Nanoindentation and Scratch Experiments

Nanoindentation, known as depth-sensitive indentation technology, is a new type of mechanical property testing method developed on the basis of the traditional Brinell hardness test and Vickers hardness test [125]. Initially, nanoindentation was used to research the mechanical properties of nanomaterials, and it was often used to detect the mechanical properties of thin films and other nanostructured materials [126]. The researchers developed a bone nanoindentation protocol to measure elastic properties consistent with macroscopic level measurement behavior. It is recommended to test large indentations with a diameter of 10 µm and depth of 500–1000 nm [139,140], leading to measured elastic moduli on the order of 10–20 GPa. Anisotropic analysis of the indentation results in two orthogonal planes showed that the moduli were consistent with the micro-tensile specimens [139]. When the same loading protocol was used for trabecular tissue, the measured elastic moduli were similar to cortical bone tissue. Low depths indentation has been used to measure the properties of individual flakes having alternating high and low moduli [141]. The bone indentation protocol is typically held under constant load for a period of time; during this time, the bone exhibits creep and stress relaxation behavior [142,143].

Similarly, nanoindentation can be used to test the mechanical properties of mineralized collagen [144,145,146]. Stanishevsky et al. prepared HA nanoparticle-collagen composites using solution deposition and electrostatic or spinning collector electrospinning and measured Young’s modulus using nanoindentation technique from 0.2 to 20 GPa and hardness from 25 to 500 MPa, depending on the composite preparation process, composition, and microstructure. When the HA content is 45–60%, the nanoindentation of Young’s modulus and hardness of the HA/collagen composite are the largest [145]. As mineralized collagens have a unique feature of composite materials in the indentation load-displacement curve and creep, and the appearance of this feature is associated with viscoelasticity, it is necessary to measure and change it to improve the mechanical properties of mineralized collagen [146].

Furthermore, the scratch test method is a high-resolution test and detection method which can observe the surface structure and morphology of materials at the microscopic scale. The test results can reveal critical surface information and mechanical parameters such as the material’s friction coefficient, hardness, and surface roughness, and combine the groove and residual morphology of the specimen surface to evaluate the friction and wear resistance of the specimen surface and the bonding ability of the film, revealing the intrinsic relationship between the material’s deep structure and its surface properties [147]. Furthermore, the scratch was used for the mechanical testing of collagen. Zhao et al. successfully calculated the critical load value of the mineralized collagen deposition coating in the scratch test [148]. The experimental results demonstrated that the critical load is proportional to the collagen concentration in the electrolyte. At high collagen concentration (500 mg/L), the critical loading of the coating was approximately twice as high as that obtained without collagen addition [148].

#### 4.2.3. AFM

In measuring the elastic modulus of collagen fibers, AFM has more sensitive detection and is less prone to make an error. AFM is an extremely versatile nanotechnology belonging to the scanning probe microscope family, and it can be used as a surface imaging tool and force sensor and actuator technology. AFM is a type of true nanoscale method where forces and deformations are on the nanometer scale. Typically, AFM is used in conjunction with other mechanical loading tools. Colin A performed the dynamic mechanical analysis of individual type I collagen fibers at low frequencies (0.1–2 Hz) using AFM (Figure 7) [133]. Different regions of procollagen have different elastic moduli. The elastic modulus of the overlap area with the highest density (approximately 5 GPa) was 160% of that of the gap area [149]. Later, AFM was used to measure and determine the flexural and shear modulus of electrospun collagen fibers. A triangular silicon nitride cantilever beam was used for vertical bending experiments. Flexural modulus dropped from 7.5716 GPa to 1.4702 GPa up to 250 nm and remained constant at 1.4 GPa for larger diameter fibers [150]. Qian et al. used AFM to record and image the nanomechanical behavior of the medullary surface of the bovine femur in situ [136].

### 4.3. Simulation Analysis Method

In response to the above experimental methods and a large number of experimental data, researchers have also established models to predict the data results. Several models to predict the mechanical properties of mineralized collagen have also been proposed. Computational models involving mostly a finite element method (FEM) and molecular dynamics (MD) atoms are briefly described in this study.

The FEM can consider the geometric details of mineralized collagen in both two-dimensional (2D) and three-dimensional (3D) space. The model can include the shape, orientation, and arrangement of various stages (Figure 8A–C). At the microscopic level, Jager proposed a geometric model for the staggered arrangement of collagen fibers and HA platelets and investigated the increase in elastic modulus and fracture stress with an increasing mineral content in the fiber [151]. Wang proposed a 2D shear lag model to explore stress concentration fields around an initial crack in a mineral-collagen composite [152]. Subsequently, some researchers began to use the cohesive FEM to analyze mineralized collagen [119]. Ana developed a 3D finite element model of staggered mineral distribution within mineralized collagen fibers to characterize the elastic behavior of lamellar bone at the submicron scale [153]. Meanwhile, a multiscale finite element framework was proposed to investigate the effect of intra–and extra-fibrillar mineralization on the elastic properties of bone tissue by considering the structural hierarchy at the nano- and micrometer scales (Figure 8D,E) [154]. The material properties and fiber network of the mineralized collagen fibers have an effect on the mechanical properties of the sub-microscale bone, according to a 3D real model of the mineralized collagen network [118]. In addition, the mechanical response of mineralized collagen at the sub-microscale is associated with the loading direction based on the different arrangements of collagen fibers.

MD simulation obtains the information and behavior of materials at the nanometer scale by studying the interaction between molecules. MD simulation can predict the overall mechanical properties of materials at the microscopic level by simulating the chemical composition and intermolecular forces of materials, and then it can also be used as the input of micromechanics or FEM. By investigating the molecular fiber toughening mechanism of mineralized collagen fibers, it was found that in a multifaceted increase in energy dissipation compared to fibers without a mineral phase [155]. Arun K. Nair investigated the mechanical properties of mineralized collagen with different mineral densities under tensile and compressive loads. Both the tensile and compressive moduli of the network increase monotonically with increasing mineral density (Figure 9B–D) [4,156]. He also investigated the effect of hydration on collagen fibers. With increasing hydration, the stress-strain behavior became more nonlinear, and the Young’s modulus of collagen fibers decreased [157]. Furthermore, the mineralized collagen fibers’ hydration has an effect on viscoelasticity. The presence of water in the fibers increases their viscosity and the energy dissipation capacity (Figure 9E–G) [158]. MD can similarly be modeled for smaller units of collagen. Computational simulations to study collagen molecular damage during cyclic fatigue loading of tendons showed that the triple-helix degeneration of collagen was positively associated with fatigue and the number of loading cycles, and the damage was associated with creep strain (Figure 9A) [159].

**Table 2 jfb-14-00212-t002:** Method for detection research mechanical properties of mineralized collagen.

Reference	Subject	Method	Detecting Parameter
Tan [133]	polycaprolactone electrospun ultrafine fiber	fiber stretching and ex situ observation	tensile malleability
Sano [132]	dentin	fiber stretching and in situ observation	bond strength
Koester [138]	bone	In situ mechanical test with SEM	mechanical properties of the longitudinal and transverse orientations of the bone
Hengsberger [139]	cortical bone of cow	nanoindentation	elastic modulus
Isaksson [142]	cortical bone of rabbit	nanoindentation	elastic modulus and viscoelastic parameters
Stanishevsky [145]	HA nanoparticle-collagen composites electrospinning	nanoindentation	Young’s modulus and hardness
Grant [149]	collagen fibrils	AFM	elastic (static) and viscous (dynamic) responses
Qian [136]	Bovine Cortical Bone	AFM	crack propagation
Jäger [151]	Bone (submicron)	FEM staggered arrangement model	elastic modulus and fracture stress
Wang [152]	Bone (submicron)	FEM 2D shear lag model	an initial crack
Vercher [153]	lamellar bone	FEM a 3D finite element model	elastic properties
Alijani [154]	lamellar bone	FEM intra and extra-fibrillar mineralization model	elastic properties
Buehler [155]	collagen microfibril	MD simulation	Young’s modulus fracture stress (mineral)
Nair [4]	collagen microfibril	MD simulation	modulus of tension (mineral)
Nair [156]	collagen microfibril	MD simulation	modulus of compression (mineral)
Milazzo [158]	collagen microfibril	MD simulation	Viscoelasticity (mineral and water content)

## 5. Conclusions

Although collagen has excellent biodegradability, low antigenicity, and biological stability, its mechanical strength is unsatisfactory. How to improve the mechanical strength of mineralized collagen is a focus of research. The combination of HA and collagen, which are also biological materials, has given great development to the bionic materials of bone. However, how to prepare bionic bone materials more in line with the natural structure of bone has become the consensus of the research community.

Admittedly, there are many maturely prepared mineralized collagens on the market, which are obtained by the following methods: 1. Precipitate the prepared collagen scaffold in the biomimetic mineralization solution. 2. Codeposition of collagen and HAp. Although this bionic bone also has the pore size and porosity of natural bone, these large pores allow cells and capillaries to grow, showing excellent bone conductivity; however, in the same proportion of components as natural bone, this bionic bone is not as good as the mechanical properties of real bone. In addition, these biomimetic bones are also quite different from natural bones at the collagen fiber level. Under the transmission electron microscope, the collagen fibers of natural bone showed obvious periodic banding (intra-fiber mineralization), while the mineralized collagen of bionic bone was only the attachment of collagen fibers and HAp (extra-fiber mineralization). In Kim’s experiment, it also showed that there were debonding particles in the artificially prepared mineralized collagen, which had a negative effect on the elastic modulus of mineralized collagen [112]. More precisely, the realization of in-fiber mineralization is also one of the effective ways to improve the mechanical properties of mineralized collagen.

In this regard, in the first section, starting from the structure of bone, we explained the multi-scale and multi-level structure of bone. From the most basic amino acids and HA crystals to mineralized collagen fibers and even to the whole bone, each layer is extremely complex, which also gives us a great challenge to understand the microstructure and mechanism of bone and to prepare artificial bionic bone. We analyzed the mechanical properties at the level of the bone unit (compact bone and cancellous bone). At the same time, stress is the main stimulus for people’s daily activities. Whether the force acts on the cells or directly on the matrix, the stress acts on the bone matrix inevitably. Mechanical stimulation is one of the influencing factors in enhancing collagen mineralization. Mineralized collagen materials are particularly important for bone, but the current research progress is far from satisfactory. Although the study confirmed at the microscopic level that fluid shear forces can regulate the rate, size, and distribution of the mineralized precursor, the implants are subjected to complex and diverse forces in the human body, and it is unknown how the complex and diverse forces regulate collagen mineralization. Bone is a piezoelectric material. Studies have shown that bone has a significant inverse piezoelectric effect on the microscopic surface. This is exciting, which helps to understand the mineralization process, and mechanical stimulation through the piezoelectric effect to produce charge affects the adhesion of HA.

In conclusion, as a bioactive material, collagen can produce different responses to different mechanical stimuli. The goal of research on bionic bone repair materials has always been to change their macroscopic and microscopic structures to have good mechanical properties similar to natural bone. The constitutive and structure-activity relationship between mechanical stimulation, mineralized structure, and mechanical properties are also worthy of further study. Mineralized collagen is expected to be better developed as a new bone repair material.

## Figures and Tables

**Figure 1 jfb-14-00212-f001:**
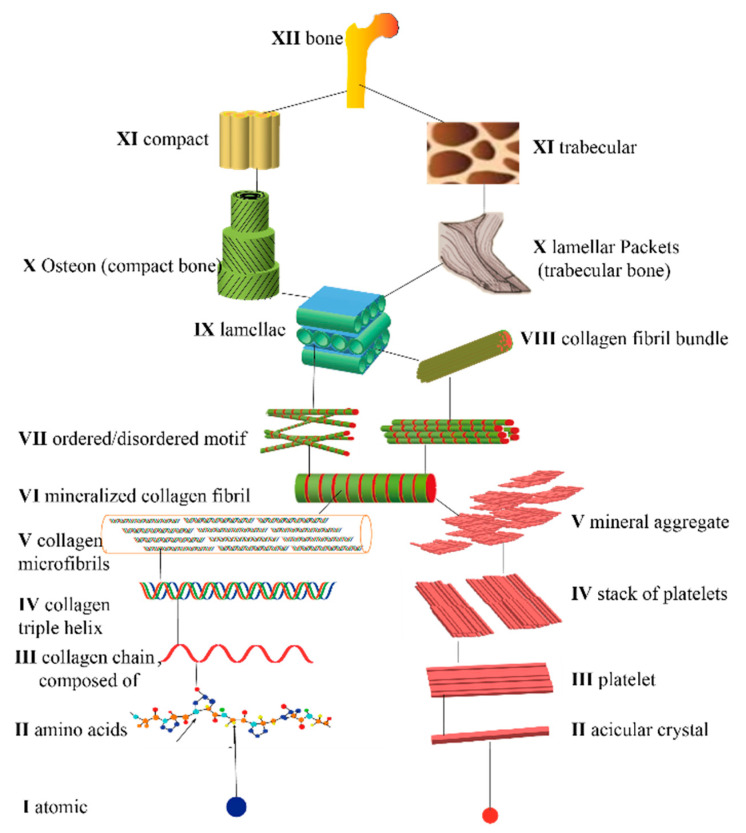
Twelve-level hierarchical structure of bone.

**Figure 3 jfb-14-00212-f003:**
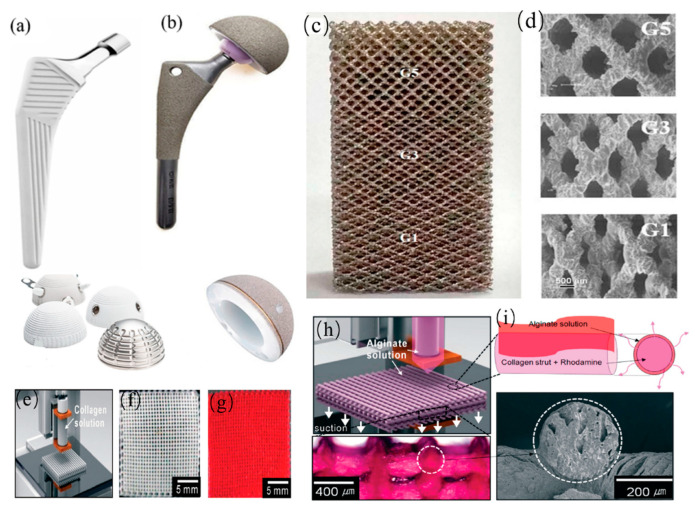
Common bone material scaffolds: (**a**) Femoral stem and articular concave scaffold; (**b**) HA-coated metal scaffold, Reprinted with permission from Ref. [46] (2023, Elsevier). (**c**,**d**) the 3D-printed vertebral body, Reprinted with permission from Ref. [52] (2023, Elsevier). (**e**–**i**) 3D alginate/collagen scaffold preparation process, Reprinted with permission from Ref. [53] (2023, ACS).

**Figure 4 jfb-14-00212-f004:**
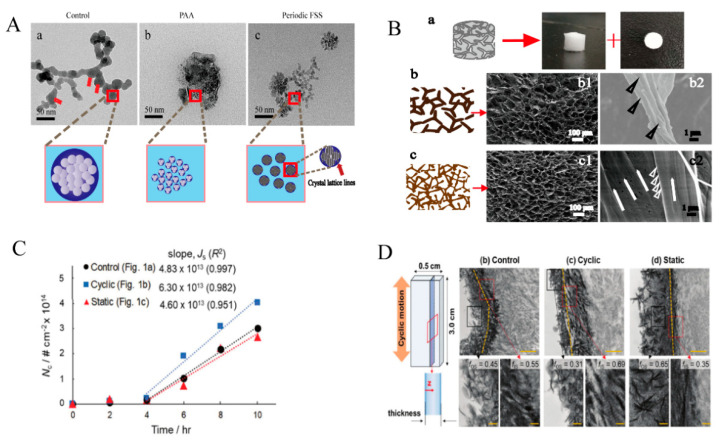
(**A**) Both PAA and periodic fluid shear stress (FSS) can control the generation of ACP Reprinted with permission from Ref. [111] (2023, RSC). (**B**) SEM images of different mineralized collagens. Figure b shows the PAA-induced normal IM collagen and figure c shows the periodic FSS and TPP-co-induced oriented HIM collagen. The SEM image (b1–b2) shows that the FSS group (Bc) is more neatly arranged than the PAA group (Bb), Reprinted with permission from Ref. [114] (2023, MDPI). (**C**,**D**) The loading cycle relative to the control variable group and the static group. The nucleation rate of mineralized collagen was higher in the stress group, Reprinted with permission from Ref. [112] (2023, RSC).

**Figure 5 jfb-14-00212-f005:**
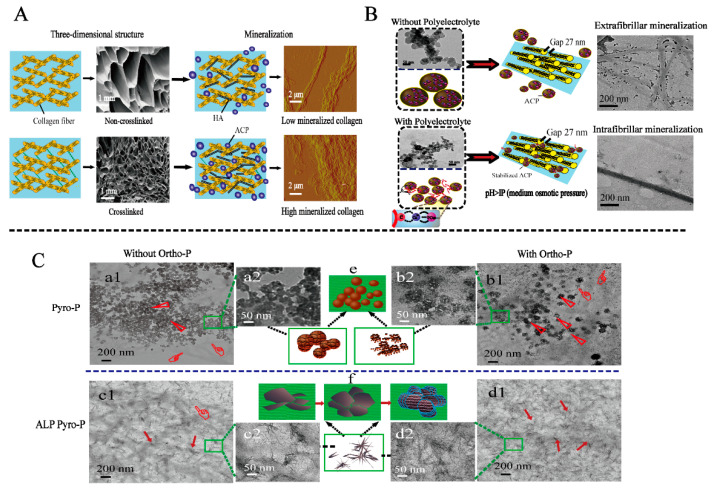
(**A**) Effect of different cross-linking degrees on collagen mineralization. With the increase of collagen cross-linking, more and more HA is attached to the surface of collagen, and collagen fibers are covered by HA (arrows), Reprinted with permission from Ref. [121] (2023, Elsevier). (**B**) The effect of polyelectrolytes on the intrafibrillar mineralization [128]. (**C**) The effect of ALP promotion on mineral crystal shape and crystallinity, c1–d2: the particles were rod-like (ALP), and a1–b2: granular particles were formed, Reprinted with permission from Ref. [128] (2023, Elsevier).

**Figure 6 jfb-14-00212-f006:**
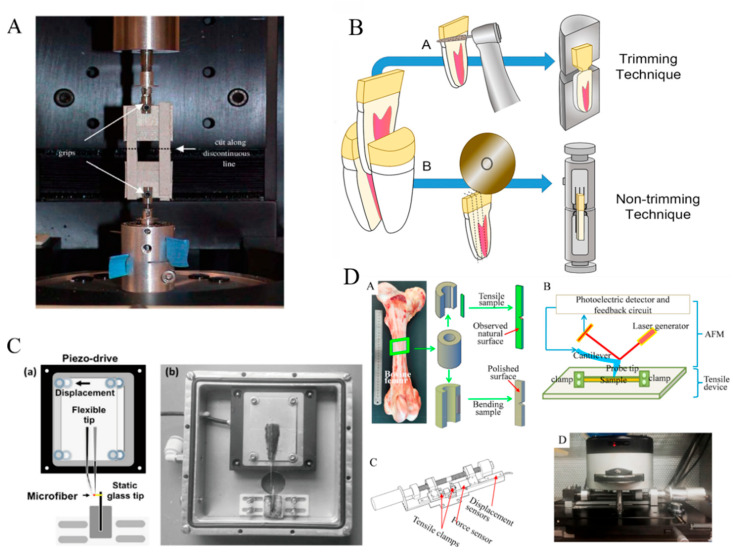
(**A**) Sample from the nano-tensile testing machine and ex situ observation, Reprinted with permission from Ref. [133] (2023, Elsevier). (**B**) In situ observation of the sample, Reprinted with permission from Ref. [132] (2023, Elsevier). (**C**) In situ stretching device for collagen fibers, Reprinted with permission from Ref. [135] (2023, Elsevier). (**D**) In situ micro-mechanical tensile mechanical test experiment by cow bone with AFM, Reprinted with permission from Ref. [136] (2023, Elsevier).

**Figure 7 jfb-14-00212-f007:**
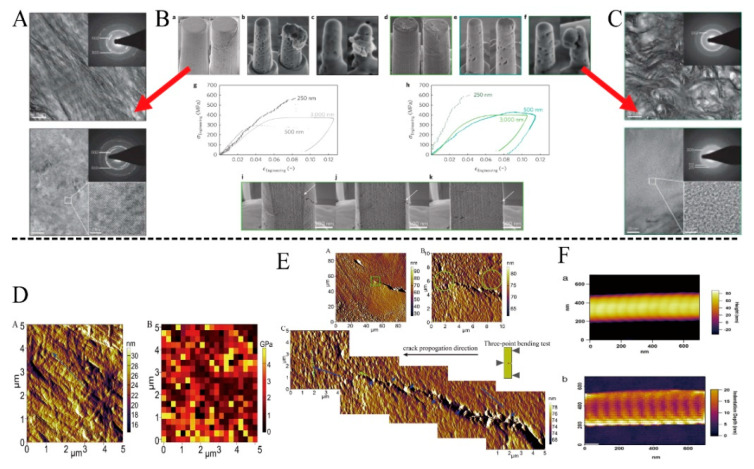
(**A**–**C**) The failure mechanism of the micron bone column under stress was observed under SEM and TEM, Reprinted with permission from Ref. [12] (2023, Springer Nature). (**B**): the failure mechanism of the ordered and disordered micron bone pillars under stress observed under SEM, (**A**), and (**C**): TEM analysis of the ordered and disordered nanostructures. (**D**) Surface nanoindentation test of compact bone under AFM and figure of stress field distribution. (a) is the AFM image of the natural surface, and the elastic modulus shown by nanoindentation in (b) [136]. (**E**) The destruction of the bone surface under stress was observed under AFM, Reprinted with permission from Ref. [136] (2023, Elsevier). (**F**) Collagen image under AFM: an image of collagen in tapping mode; b, indentation data of collagen, Reprinted with permission from Ref. [149] (2023, Elsevier).

**Figure 8 jfb-14-00212-f008:**
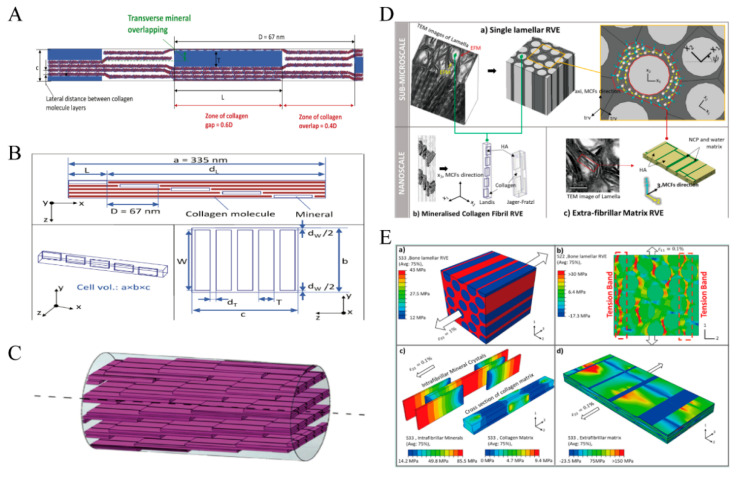
Finite element simulation. (**A**–**C**) Collagen and minerals are interlaced within the fibers, Reprinted with permission from Ref. [153] (2023, Elsevier). (**D**,**E**) (**D**): Structural modeling of bone at the micro and nano-scales; (**E**): Stress cloud diagram of the model at 0.1% strain. a: Axial stress distribution of MCFs at 0.1% strain; b, c: Stress distribution in the lamellar structure of bone; d: Stress distribution of extra-fibrillar matrix, Reprinted with permission from Ref. [154] (2023, Elsevier).

**Figure 9 jfb-14-00212-f009:**
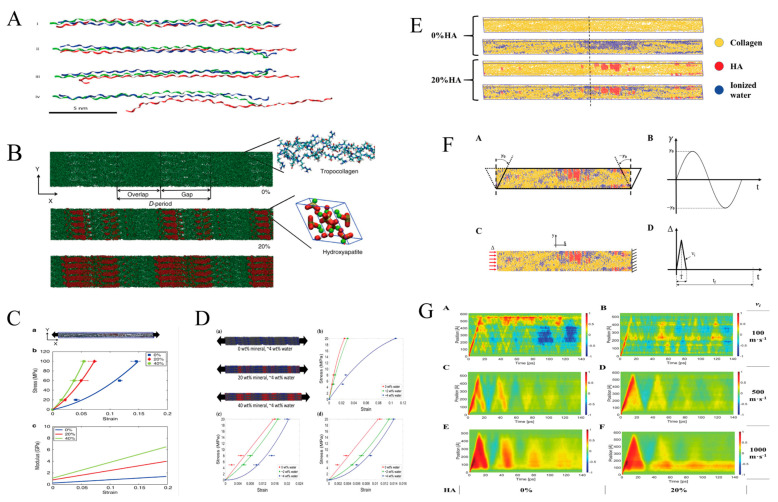
Molecular dynamics simulation. (**A**) The triple helical peptide chain undergoes unwinding under cyclic fatigue loading and thus breaks, Reprinted with permission from Ref. [159] (2023, AAAS). (**B**) Collagen microfibril model with 0%, 20%, and 40% mineralization simulation modeling of mineralized collagen, Reprinted with permission from Ref. [4] (2023, Springer Nature). (**C**,**D**) Stretching and compression models of collagen microfibrils with different mineralization rates Reprinted with permission from Refs. [4,156] (2023, Springer Nature). (**E**–**G**) Study on the viscoelastic behavior of mineralized collagen by mineral and water content, Reprinted with permission from Ref. [158] (2023, RSC).

## Data Availability

There are no additional data available for this study other than what is reported in the manuscript.

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
