# Peer review of "Biomechanical Characteristics and Analysis Approaches of Bone and Bone Substitute Materials"

_jfb, 2023, doi:10.3390/jfb14040212_

Round 1

Reviewer 1 Report

This article is a review on the anatomy, composition and mechanics of the bone tissue, and biomimetic bone materials developed for bone repair. The article also describes different methods employed to improve the biomechanical properties of mineralized collagen materials as well as characterize the biomechanical properties of these materials. While the paper has the potential to be a good contribution to the scientific literature, there are various concerns that must be addressed:

1. English and grammar must be carefully revised throughout the article.

2. Table 1 is confusing in the way it is laid out with overlapping rows. A more clear depiction of the table is needed.

3. Abstract is somewhat misleading as it is written as if this is a research article. This reviewer suggests that the abstract be rewritten to highlight the focus of the review paper more clearly.

4. Addition of more tables (e.g., different methods to make mineralized collagen materials and the measured biomechanical properties reported for these materials) will enrich the review paper and be helpful to the reader.

5. In section 3.1, description of collagen alignment (Ref 92) seems misplaced. What is the impact of collagen alignment on mineralization? This warrants a separate section.

6. A number of methods (e.g., electrochemical alignment, plastic compression) are reported in the literature to make composite/mineralized collagen materials. These are not described in this review. The reviewer recommends the authors to look at the following references: 1) Marelli, Benedetto, et al. "Three-dimensional mineralization of dense nanofibrillar collagen− bioglass hybrid scaffolds." Biomacromolecules 11.6 (2010): 1470-1479. 2) Nijsure, Madhura P., et al. "Bioglass incorporation improves mechanical properties and enhances cell‐mediated mineralization on electrochemically aligned collagen threads." Journal of biomedical materials research Part A 105.9 (2017): 2429-2440.

There are many more such articles that focus on incorporating ceramic materials in collagen to promote mineralization. A separate section on this topic will strengthen the review.

7. Conclusions and future directions section is very weak and does not clearly describe the state of the field. Why are biomimetic materials mechanically weaker than native bone tissue? What are suggested directions to further improve the biomechanical properties of these materials?

Reviewer 2 Report

Comprehensive information has been given on the HAp which is the main component of the bones and the study is focused on improving and analyzing the mechanical strength of mineralized collagen. To improve the mechanical properties of mineralized collagen, this study introduced the structure and mechanical properties of native bone. 

The work can be accepted for publication in the journal but it seems some information (maybe one paragraph) is missing about the doping to the HAp to improve its ability. I suggest the following and some other references about the doping studies of HAp:

1-  https://doi.org/10.1016/j.molstruc.2023.135095

2- https://doi.org/10.3390/ma15207211

Reviewer 3 Report

This review paper is well organized. The contents provide much useful information on bone and bone substitute materials. It is well to be published.

Reviewer 4 Report

This review reports Biomechanical Characteristics and Analysis Approaches of 2 Bone and Bone Substitute Materials.

While the work is overall carried out well and the review support the conclusion, there are several issues that need attention and upon addressing those issues the review can be accepted in jfb

Title and abstract:

1.     Firstly, there are numerous typos (overtyping) throughout the manuscript, all requiring attention (the abstract has such errors).  There are several grammatical errors that needed to be corrected. I urge the authors to thoroughly go through the entire manuscript and check every line for spelling, grammar, or sentence construction-related errors as without these measures the account is unreadable.

1-    In manuscript: explain each word for first time in the begging and used its abbreviation after that [eg: amorphous calcium phosphate (ACP)]

2-     The authors are suggested to add the descriptions on novelty, significance, and in-depth discussion.

3-    Proper discussion and conclusion outcome of the results to be presented in the manuscript with supporting references.

4-    The review need to detect plagiarism and English editing

Round 2

Reviewer 1 Report

Concerns addressed satisfactorily. 

Reviewer 4 Report

Dear editor,

The authors reply to all comments.

best regards

Sanaa